# Design, Synthesis, and In Vitro and In Vivo Antifungal Activity of Novel Triazoles Containing Phenylethynyl Pyrazole Side Chains

**DOI:** 10.3390/molecules27113370

**Published:** 2022-05-24

**Authors:** Tingjunhong Ni, Zichao Ding, Fei Xie, Yumeng Hao, Junhe Bao, Jingxiang Zhang, Shichong Yu, Yuanying Jiang, Dazhi Zhang

**Affiliations:** 1Department of Pharmacy, Shanghai Tenth People’s Hospital, School of Medicine, Tongji University, No. 1239 Siping Road, Shanghai 200092, China; ntjh860415@126.com; 2Department of Organic Chemistry, School of Pharmacy, Navy Medical University, No. 325 Guohe Road, Shanghai 200433, China; dingzc_pharm@126.com (Z.D.); xiefei3032011003@smmu.edu.cn (F.X.); haoyumenggogo@163.com (Y.H.); baojunhe021@smmu.edu.cn (J.B.); 3Center for New Drug Research, School of Pharmacy, Navy Medical University, No. 325 Guohe Road, Shanghai 200433, China; zhangjingxiang@smmu.edu.cn

**Keywords:** triazole, CYP51, antifungal, molecular docking, synthesis

## Abstract

A series of triazole derivatives containing phenylethynyl pyrazole moiety as side chain were designed, synthesized, and most of them exhibited good in vitro antifungal activities. Especially, compounds **5k** and **6c** showed excellent in vitro activities against *C. albicans* (MIC = 0.125, 0.0625 μg/mL), *C. neoformans* (MIC = 0.125, 0.0625 μg/mL), and *A. fumigatus* (MIC = 8.0, 4.0 μg/mL). Compound **6c** also exerted superior activity to compound **5k** and fluconazole in inhibiting hyphae growth of *C. albicans* and inhibiting drug-resistant strains of *C. albicans*, and it could reduce fungal burdens in mice kidney at a dosage of 1.0 mg/kg. An in vivo efficacy evaluation indicated that **6c** could effectively protect mice models from *C. albicans* infection at doses of 0.5, 1.0, and 2.0 mg/kg. These results suggested that compound **6c** deserves further investigation.

## 1. Introduction

During the last four decades, the morbidity of invasive fungal infections (IFIs) has been increasing in patients which acquired immunodeficiency, and the mortality keeps high [1,2]. According to the clinic report, *Candida albicans*, *Cryptococcus neoformans* and *Aspergillus fumigatus* are the three most common human pathogenic fungi [3,4]. It is estimated that more than 300 million people suffer from serious fungal-related diseases, and fungi collectively kill over 1.6 million people annually, which is more severe than malaria and similar to tuberculosis [5]. Moreover, the rate of emergence of drug resistance is greater than the discovery pace of antifungal research. Therefore, it is urgent to develop novel antifungal agents [6].

Compared with the other few antifungal agents, including amphotericin B, flucytosine and echinocandins, azole antifungals, especially triazoles, were widely applied in treating IFIs and were much more concerned in research due to their higher therapeutic index, broader-spectrum, lower toxicity and superior druggability [7,8,9]. However, their extensive use has led to the occurrence of resistant fungi, which may cause the failure of antifungal treatment and greatly limit therapeutic options [10]. Hence, new antifungal azoles are therefore highly needed.

Azoles antifungal agents act by inhibition of fungal lanosterol 14α-demethylase (CYP51), which is a necessary enzyme for catalyzing the oxidative removal of the 14α-methyl in sterol biosynthesis of fungi [11]. By summarizing the structures of triazole antifungals approved for the treatment of invasive fungal infections, as shown in Figure 1, a clear pharmacophore can be concluded to possess a triazole and a halophenyl ring, which have been proved to bind the heme iron and the hydrophobic pocket of CYP51 [12]. The side chain in the right part of each structure, shown in Figure 1, occupies the substrate access channel of CYP51. Structurally, the difference between newer azole antifungals mainly focuses on the type of side chain attached to the carbinol center of the triazole alcohol scaffold. Accordingly, most of the recent efforts aimed to optimize this part of the molecule correspondingly. Therefore, the study of antifungal azoles has been mainly focused on the structural optimization and SAR research on the side chain for new drugs [13].

Guided by the binding modes of triazole antifungal agents, a number of new triazoles were rationally designed and synthesized by our group [14,15,16,17,18]. Among them, a series of triazoles containing alkynyl side chains showed good antifungal activity with a broad spectrum [18]. They exhibited superior activity to fluconazole and comparable activity to voriconazole, and specially compound **A5** (Figure 2) was subjected to evaluate its in vivo efficacy as a promising compound. Unfortunately, it possessed inferior activity in the murine model of disseminated *C. albicans* infection. Considering the lack of hydrogen bond donors and acceptors as well as the number of rotatable bonds in compound **A5**, further lead optimization was focused on ameliorating drug-likeness in order to improve the in vivo potency. Herein, a series of novel side chains of triazole antifungals were designed by hybridizing the side chains in lead compound **A5** and fluconazole and replacing the 1,2,4-triazole with pyrazole based on the bioisosterism principle [19] to construct a phenylethynyl pyrazole side chain in Figure 2. Herein, we reported in vitro antifungal activity and SAR of all the target compounds **5a–v** and **6a–e**, and in vivo antifungal potency of **6c**.

## 2. Results and Discussion

### 2.1. Chemistry

As depicted in Figure 1, the target compounds were synthesized, starting from 1-((2-(2,4-difluorophenyl)oxiran-2-yl)methyl)-1*H*-1,2,4-triazole methane sulfonate (**1**), which the opening-ring reacted with 4-iodo-1*H*-pyrazole in the presence of K_2_CO_3_ in DMF. Sonogashira reaction of the key intermediate (**2**) with substituted (4-ethynylphenoxy)methyls or aryl alkyne was conducted in an NMP solution of Pd(PPh_3_)_2_Cl_2_, CuI and DIEA at 60 °C to afford (**3**) or **6a–e**. In addition, the ester hydrolysis reaction of methyl 4-((1-(2-(2,4-difluorophenyl)-2-hydroxy-3-(1*H*-1,2,4-triazol-1-yl)propyl)-1*H*-pyrazol-4-yl)ethynyl)benzoate (**4**) was reacted with LiOH in a mixed solvent of H_2_O and THF, which was then subjected to the amidation reaction conditions with substituted alkyl amine or aromatic amine to give target compounds **5a–5v**.

### 2.2. In Vitro Antifungal Activity

All 29 compounds were evaluated by means of the minimum inhibitory concentration (MIC) according to the regulations recommended by the National Committee for Clinical Laboratory Standards (NCCLS) [20,21]. MIC was defined as the first well with an approximate 80% reduction in growth compared to the growth of the drug-free well. The three most common human pathogenic fungi, including *Candida albicans*, *Cryptococcus neoformans* and *Aspergillus fumigatus*, were tested. Clinic antifungal drugs fluconazole (FCZ) and posaconazole (POS) were used as the reference drugs to compare with target compounds. All data were the means of three replicate tests performed with each target compound and are summarized in Table 1.

In this study, the grades of MIC values were regarded as: excellent: <0.25 μg/mL; good: 0.25–1 μg/mL; moderate: 1–64 μg/mL. As shown in Table 1. Most of the target compounds exhibited good to excellent inhibitory activity against *C. albicans* and *C. neoformans* with MIC values ranging from 1 to 0.0625 μg/mL. Especially, compounds **5a**, **5b**, **5j, 5k, 5o**, **6a** and **6c** exhibited MIC values of 0.0625 μg/mL against *C. albicans*. Compounds **6a** and **6c** exhibited MIC values of 0.0625 μg/mL against *C. neoformans*. These compounds showed superior activity to fluconazole (FCZ) and posaconazole (POS). Moreover, compounds **5k**, **5t**, **5u**, **5v**, **6c** exhibited moderate activity against *A. fumigatus* (MIC = 8.0 μg/mL). However, fluconazole (FCZ) is inactive against *A. fumigatus* (MIC > 64.0 μg/mL). Among all the compounds, **5k** and **6c** are promising leads for the development of new generations of triazole antifungal agents. These results suggested that phenylethynyl pyrazole could be considered a novel privileged structure of side chain that deserves further investigation.

Preliminary structure-activity relationships (SARs) of compounds **5a–i** with alkyl-substituted amides as side chains shorter or longer in length and smaller or bulkier in size, were investigated. Smaller and shorter side chains in compounds **5a–i** are favorable to the antifungal activity against *C. albicans* and *C. neoformans*. For example, the MIC values of compounds **5a** and **5b**, which bear methyl and isopropyl, respectively, were better than that of the other compounds, FCZ, and POS against *C. albicans* and *C. neoformans*. The main problem of the design of compounds **5a–i** with substituents R as aliphatic groups is that all compounds exhibited inactive against *A. fumigatus*.

Compounds **5j–v,** as derivatives of **5a** with one more aromatic ring introduced to the side chain, showed good in vitro antifungal activity. Especially, compounds **5k**, **5o**, **5t**, **5u**, and **5v** exhibited excellent activities against *C. albicans* (MIC = 0.0625~0.25 μg/mL), *C. neoformans* (MIC = 0.125~0.25 μg/mL), and moderate activities against *A. fumigatus* (MIC = 8.0~16.0 μg/mL). Furthermore, compounds **6a**–**e**, designed by converting the amide side chain of **5a–v** into an ether bond-linked side chain, exerted very good antifungal activity. Compound **6c** showed the most potent activity against *C. albicans* (MIC = 0.0625 μg/mL), *C. neoformans* (MIC = 0.0625 μg/mL), and *A. fumigatus* (MIC = 4.0 μg/mL).

Based on the results of the in vitro antifungal activity of fluconazole-sensitive strains, the active compounds **5a**, **5b**, **5j**, **5k**, **6a** and **6c** were further evaluated against fluconazole-resistant strains of *C. albicans* 100 and 103. As summarized in Table 2, the compounds showed moderate antifungal activities against fluconazole-resistant strains of *C. albicans* with MIC values in the range of 2.0 to 16.0 µg/mL. Compound **6c** also exhibited higher activity than other compounds against both tested drug-resistant strains with MIC values of 4.0 µg/mL.

### 2.3. Theoretical Evaluation of ADME/T Properties

Since compounds **5a**, **5b**, **5j**, **5k**, **6a** and **6c** exhibited excellent antifungal activity, their ADMET prediction was performed using the DS-ADMET and DS-TOPKAT modules to evaluate their druggability, and the predicted data are summarized in Figure 3 and Appendix A.

Most compounds were positioned in the 95% and 99% confidence ellipses for human intestinal absorption (absorption), and only compound **6c** was positioned between the interval of 95% and 99% confidence ellipses (Figure 3). Meanwhile, compounds **5a** and **5b** were positioned between the interval of 95% and 99% confidence ellipses for blood–brain barrier (BBB) penetration, and compounds **5j**, **5k**, **6a** and **6c** were beyond the 99% confidence ellipse of the BBB model. These results indicated that these compounds with proper aqueous solubility and relatively low BBB penetration may enter blood circulation through intestinal absorption and exert their antifungal effect in vivo without causing nervous system damage.

Next, we analyzed the A logP and PSA of these compounds, and their values were distributed within a reasonable absorption range, suggesting that these compounds can achieve appropriate drug concentrations in vivo (Appendix A). In addition, the toxicological properties of target compounds were further predicted according to the TOPKAT calculation module. We found these compounds had the characteristics of non-mutagen, non-irritant, and non-carcinogen, which greatly reduces the drug risk. Although all compounds showed strong skin sensitization, we did not consider this side effect as a concern, as the control drug fluconazole had the same predicted results.

### 2.4. Anti-Hyphal Activity

Hyphae growth is a significant morphological feature of fungi and is one of the virulent factors [22]. Morphological transitions from yeast to filamentous forms are the major contributor to the in vivo pathogenicity of *C. albicans* [23,24,25,26]. As the basis for studying in vivo potency, we further investigated the activity of compounds **5k** and **6c** against the yeast-to-hyphae transition of *C. albicans* with fluconazole as the control drug. As shown in Figure 4, compounds **5k** and **6c** exhibited mild activity against fungi hyphal formation, with fewer hyphae and more pseudohyphal cells compared with the group without drug treatment at 1.0 μg/mL or higher concentrations. Even at the concentration of 0.0625 μg/mL, compound **6c** showed an obvious difference between compound **5k** and fluconazole, which grew more hyphaes.

### 2.5. Fungal Burden Evaluation

*C. albicans* has a strong tropism in kidney tissue, so the fungal burden is an important indicator for evaluating systemic fungal infection. Compound **6c**, which demonstrated significant in vitro and anti-hyphal activity, was tested for the evaluation of the fungal burden of systemic *C. albicans* SC5314 in ICR mice. Following 3 days of treatment, we determined the changes in the fungal burdens in the kidney by measuring the number of CFU in Figure 5. Significant reductions in fungal burdens were observed with compound **6c** in a dose-dependent manner compared with the vehicle control. Meanwhile, there is no significant difference between compound **6c** (at the dosage of 1.0 mg/kg) and FCZ (at the dosage of 0.3 mg/kg). The results highlighted the antifungal potential of compound **6c**.

### 2.6. In Vivo Potency

Considering the good evaluation of the fungal burden of compound **6c**, an in vivo potency study was evaluated in disseminated *C. albicans* SC5314 models. As summarized in Figure 6, interestingly, compound **6c** showed potent in vivo antifungal activity. The survival rate of compound **6c** exhibited a dose-dependent manner. At the dose of 0.5 mg/kg, 20% of ICR mice survived at the end of the test, which was moderate to that of the fluconazole group (0.5 mg/kg). Meanwhile, it could effectively protect mice from fungal infection at the dose of 2.0 mg/kg (*p* < 0.001). This result indicated that it possessed potent activity, which could effectively protect mice from *C. albicans* infection.

### 2.7. Molecular Docking

In order to investigate whether the target compounds could have a well binding with CYP51, a molecular docking study was performed. All compounds were docked into the active site and scored using the Surflex-Dock program in the SYBYL-X 2.0 software (Appendix A). The published crystal structures of *C. albicans* CYP51 (PDB ID: 5TZ1) served as a useful template for generating binding modes [27]. The most active compound, **6c,** will be shown in Figure 7 as a representative. As shown in Figure 7, the iron in the heme group was coordinated by triazole moiety, meanwhile, the 2,4-difluorophenyl could be placed into the hydrophobic pocket formed by Tyr-132, Phe-126, Met-306 and Phe-145. As surmised, the rigid linear phenylethynyl moiety was designed as a bar to anchor the side chain into the hydrophobic channel. The hydroxyl group in compound **6c** formed a hydrogen-bonding interaction with Tyr-132 (2.4 Å). The long alkynyl side chain extended into a hydrophobic channel formed by the surrounding residues Tyr-118, Phe-228, Leu-376 and Phe-380 to form *van der Waals* and hydrophobic interactions. It is worth noting that the π–π stacking interaction was found between 2,4-difluorophenyl and Phe-126 and between pyrazole and Tyr-118, respectively. This may further improve the affinity and specificity of the inhibitors. Due to flexibility, the benzyloxy produced a bend in the channel. Especially, the strong hydrogen bonds existing between terminal nitrile and Lys-90 (1.9 Å) and between benzyloxy and His-377 (3.0 Å) may be a significant factor in the effective antifungal activity.

## 3. Materials and Methods

### 3.1. Chemistry

^1^H and ^13^C nuclear magnetic resonance (NMR) spectra were reported in DMSO-*d_6_* unless otherwise indicated, by a Bruker AC-300P spectrometer. Tetramethylsilane (TMS) was considered as the internal standard. Chemical shifts (*δ* values) and coupling constants (*J* values) are given in ppm and Hz, respectively. HPLC purity was determined by Agilent Technologies 6120 Quadrupole LC-MS. HRMS analyses were produced on an Agilent Technologies 6538 UHD Accurate-Mass Q-TOF LC/MS. Silica gel plates GF254 (Yantai Huanghai Chemical, Yantai, China) were applied to thin-layer chromatography (TLC) analysis. All the solvents and reagents were purchased from commercial vendors and were used as received or dried prior to use as needed.

#### 3.1.1. Procedure for the Synthesis of 2-(2,4-Difluorophenyl)-1-(4-iodo-1*H*-pyrazol-1-yl)-3-(1*H*-1,2,4-triazol-1-yl)propan-2-ol (**2**)

To a solution of compound **1** (50 mmol) and 4-Iodo-1*H*-pyrazole (50 mmol) in DMF (150 mL) was added K_2_CO_3_ (100 mmol). The mixture was stirred continuously for 4 h and heated at 80 °C. The reaction was monitored by TLC. After the reaction was finished, the mixture was cooled to room temperature, poured into ice water, then stirred for 1 h. The product solid was filtered and then dried at 50 °C to obtain compound **2** (15.5 g, yellow solid, yield 72%).

#### 3.1.2. Procedure for the Synthesis of Methyl 4-((1-(2-(2,4-Difluorophenyl)-2-hydroxy-3-(1*H*-1,2,4-triazol-1-yl)propyl)-1*H*-pyrazol-4-yl)ethynyl)benzoate (**3**)

Under nitrogen atmosphere, compound **2** (40 mmol) and Methyl 4-ethynylbenzoate (40 mmol) were dissolved in NMP (140 mL). To this solution was added CuI (20 mmol%), Pd(PPh_3_)_2_Cl_2_ (5 mmol%) and DIEA (200 mmol). The mixture was degassed under a nitrogen atmosphere prior to heating at 60 °C for 6 h. The reaction was monitored by TLC. After the reaction was finished, the mixture was poured into ice water, and then extracted with ethyl acetate (3 × 250 mL). The organic phases were combined, washed with saturated aqueous sodium chloride solution (2 × 300 mL), dried over anhydrous Na_2_SO_4_ and evaporated under reduced pressure. The crude products were purified by chromatography on silica gel (PE:EA = 20:1~5:1) to obtain compound **3** (14.57 g, yellow solid, yield 78%).

#### 3.1.3. Procedure for the Synthesis of 4-((1-(2-(2,4-Difluorophenyl)-2-hydroxy-3-(1*H*-1,2,4-triazol-1-yl)propyl)-1*H*-pyrazol-4-yl)ethynyl)benzoic acid (**4**)

To a solution of compound **3** (31 mmol) in a mixed solution of THF (100 mL) and H_2_O (100 mL), were added LiOH (200 mmol). The mixture was stirred continuously for 6 h and heated at 50 °C. The reaction was monitored by TLC. After the reaction was finished, THF was evaporated under reduced pressure. Aqueous hydrochloric acid (5 mol/L) was dropped into the remaining liquid to adjust pH to 3–4. After stirring for 1 h, the precipitated solid was filtered and dried to obtain compound **4** (12.1 g, yellow solid, yield 67%).

#### 3.1.4. General Procedure for the Synthesis of Target Compound (**5a–5v**)

To a solution of compound **4** (1.0 mmol) and amines (1.0 mmol) in DMF (5 mL), were added DIEA (2.0 mmol) and PyBOP (1.1 mmol). The mixture was stirred continuously for 4–8 h and heated at 50 °C. The reaction was monitored by TLC. After the reaction was finished, the mixture was poured into ice water, and then extracted with ethyl acetate (3 × 20 mL). The organic phases were combined, washed with saturated aqueous sodium chloride solution (2 × 30 mL), dried over anhydrous Na_2_SO_4_ and evaporated under reduced pressure. The crude products were purified by reverse phase (mobile phase was A: H_2_O, B: acetonitrile, gradient elution, 0–20 min, 30–90% B) and then lyophilized to obtain the target compounds.

#### 3.1.5. General Procedure for the Synthesis of Target Compound (**6a–6e**)

Under nitrogen atmosphere, compound **2** (1.0 mmol) and alkynes (1.0 mmol) were dissolved in NMP (10 mL). To this solution was added CuI (20 mmol%), Pd(PPh_3_)_2_Cl_2_ (5 mmol%) and DIEA (5 mmol). The mixture was degassed under nitrogen prior to heating at 60 °C for 6 h. The reaction was monitored by TLC. After the reaction was finished, the mixture was poured into ice water, and then extracted with ethyl acetate (3 × 20 mL). The organic phases were combined, washed with saturated aqueous sodium chloride solution (2 × 30 mL), dried over anhydrous Na_2_SO_4_ and evaporated under reduced pressure. The crude products were purified by chromatography on silica gel (PE:EA = 10:1~5:1).

### 3.2. In Vitro Antifungal Activities Assay

In vitro antifungal activity was measured according to the protocols from the National Committee for Clinical Laboratory Standards (NCCLS) [20,21]. The serial dilution method in a 96-well microtest plate was used to measure the minimum inhibitory concentration (MIC) of the target compounds. For *C. albicans* and *C. neoformans*, the initial concentration of fungal suspension in the RPMI 1640 medium was 10^3^ CFU/mL. For *A. fumigatus*, the initial concentration of fungal suspension in the RPMI 1640 medium was 5 × 10^3^ CFU/mL. Targeted compounds were dissolved in DMSO and serially diluted in a growth medium. The final concentrations of each well ranged from 0.125 to 64 µg/mL. The yeasts were incubated at 35 °C and the filamentous fungi were incubated at 37 °C. After 48 h incubation, the optical density (OD630 nm) in each well was measured by spectrophotometer. The MICs were defined as the minimum concentration of drugs to inhibit ≥80% growth of fungal cells compared to that of a drug-free control at 30 °C at 48 h incubation.

### 3.3. Hyphal Formation Assay

Firstly, *C. albicans* SC5314 cells were harvested by centrifugation (3000 rpm, 5 min) and washed with PBS three times. The *C. albicans* suspension was then adjusted to 1 × 10^6^ cells/mL with Spider medium. The *C. albicans* suspension was divided into every well in 6-well plates with different concentrations of compounds **5k**/**6c** added. Then the 6-well plates were incubated at 37 °C. After 3 h incubation, the cellular morphology was photographed.

### 3.4. Fungal Burden Evaluation

In the fungal burden study, antifungal treatments began 1 day after fungal inoculation and continued for 3 days. The day after therapy had stopped, mice were humanely euthanized and the kidneys were collected for the quantitative determination of the tissue fungal burden. After the weights of kidneys were determined, tissues were homogenized in 1 mL PBS. Homogenates were serially diluted in 10-fold steps and aliquots (100 μL) of homogenate were plated on Sabouraud dextrose agar (SDA) plates. The plates were incubated at 30 °C for 72 h, and the numbers of CFU were counted. The fungal burdens were indicated as Log_10_CFU/g.

### 3.5. In Vivo Antifungal Potency

All animal experiments were done according to institutional guidelines and were approved by the Institutional Animal Care and Use Committee (IACUC) of Second Military Medical University. Female ICR mice (weighing between 18 to 22 g) were housed and fed to acclimatize for 3 days. All mice were operated on intraperitoneal injection with 0.4 mL of cyclophosphamide (100 mg/kg, 3 days, qd). Then, mice were administered orally with 0.2 mL of a suspension containing 5 × 10^6^ CFU/mL of *C. albicans* SC5314. In this model, compound **6c** and FCZ (as a suspension in 0.5% carboxymethylcellulose in distilled water) were administered orally 2 h after infection fungi (7 days, qd). The control group consisted of mice treated with NS. Mice were monitored and recorded for survival conditions once daily for a total period of 20 days post-infection. At the end of the observation period, the surviving mice were humanely sacrificed.

### 3.6. Statistics

Survival was plotted by Kaplan–Meier analysis, and a log-rank test was used to assess for significant differences in median survival time. Differences in fungal burdens between groups were assessed for significance by analysis of variance (ANOVA) with Tukey’s posttest for multiple comparisons. A *p*-value of <0.05 was considered statistically significant, a *p*-value of <0.01 was considered statistically highly significant, and a *p*-value of <0.001 was considered statistically extremely significant.

### 3.7. Computational Methodology

#### 3.7.1. ADME/T Prediction

All target compounds were predicted using DS-ADMET and DS-TOPKAT modules. The operation process was performed as follows: the target compound files were imported into the program, and “ADMET descriptors” and “Toxicity Prediction” modules were selected, respectively. The prediction items (aqueous solubility, blood brain barrier penetration, intestinal absorption, plasma protein binding, FDA rodent carcinogenicity, Ames mutagenicity, Skin_Irritancy, Skin_sensitization) were set as research objects in the parameter browser, respectively. Finally, the program was run to obtain the corresponding result.

#### 3.7.2. Molecular Docking

The structure of compound **6c** was drawn by SYBYL-X 2.0 software, optimized using the standard Tripos molecular mechanics force field with a Gasteiger–Hückel charge, and other parameters were set as default values. Import the PDB (5TZ1) file into SYBYL-X 2.0, first remove water molecules and the ligand, then analyze the protein structure, repair the terminal side chains, and add hydrogen to amino acid residues. The protein structure was optimized using the standard Tripos molecular mechanics force field with the AMBER7 FF99 charge. Next, the active pocket was generated, and molecule **6c** was docked into the active pocket through the “Surface Dock” module. Finally, this module will score the interaction between **6c** and CYP51 protein and retain the 20 highest scoring ligand-protein complex structures for analysis. The docking results were plotted by Pymol software.

## 4. Conclusions

In summary, a series of novel triazole derivatives containing alkynyl side chains have been designed, synthesized, and their antifungal activities were evaluated for the three most common human pathogenic fungi. Most of the target compounds have strong antifungal activities against the tested fungi, including fluconazole-resistant strains, and especially, compound **6c** exhibited potent in vitro antifungal activity against the three tested fungi strains and in vivo efficacy in the mice model of disseminated *C. albicans* infection. It could effectively protect mice models from *C. albicans* infection at doses of 0.5, 1.0, and 2.0 mg/kg. A molecular docking study also demonstrated that the long alkynyl side chain could form closely the *van der Waals* and hydrophobic interactions with CYP51. The strong hydrogen bond existing between 4-cyanobenzyloxy and residues played an important role in the binding affinity and antifungal activity. Further pharmacokinetic evaluation and structure optimization on compound **6c** is under investigation.

## Data Availability

Not applicable.

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
