# Peer review of "Design, Synthesis, and In Vitro and In Vivo Antifungal Activity of Novel Triazoles Containing Phenylethynyl Pyrazole Side Chains"

_molecules, 2022, doi:10.3390/molecules27113370_

Round 1
Reviewer 1 Report
The manuscript of Ni et al. describes the synthesis of new triazoles containing phenylethynyl pyrazole as side chain and the in vitro and in vivo evaluation of antifungal activity.
The results obtained are interesting. The paper is well written, concise and the conclusions are consistent. The scientific rationale is well explained.
There are only two typographical errors:
Figure 3: not "1,2,4-trizole" but "1,2,4-triazole"
Line 91: not "6h ;." but "6h."
Finally in line 256: the crude products were purified by reverse phase. The stationary phase and mobile phase should be specified.
In my opinion the manuscript is worthy of publication.
Reviewer 2 Report
This is a well-focused work aimed at the development and in-vitro and in-vivo evaluation of new antifungal triazoles as chemotherapeutic agents against the most common IFIs in humans. The molecular design used is structurally reasonable and is supported by other structures with antifungal activity already reported by the authors. Extensive underlying laboratory work is reflected with the synthesis, molecular characterization, and antifungal in-vitro activity evaluation of nearly 30 new fluconazole derivatives. The developed ligands were initially tested against strains sensitive to the reference drugs, and then the best candidates were selected for testing against resistant strains. Additional in-vitro assays of anti-hyphal activity support the selection of the best candidate (compound 6c) to proceed to the in-vivo assay phase. Evaluation of fungal burden in kidney tissue and studies of antifungal potency in mice agree that compound 6c has a high potential as a nucleus for the development of new triazoles for the treatment of C. albicans infections.
The results presented are of great interest to the scientific community involved in the search for new ligands with effective antifungal activity, as well as for organic chemists working with triazoles, however I highlight below some aspects that I consider should be covered first to guarantee a work of sufficient quality to be published in the journal Molecules.
- At the end of the abstract text, the conduct of a SAR study on compound 6c is mentioned, while such a study is not demonstrated in the body of the document. Perhaps what the authors are really referring to here is the development of a molecular docking study on compound 6c. This needs to be clarified.
- With the CYP51 protein prepared and the molecular docking parameters already defined in the Sybyl software, it would have been relatively straightforward to perform a comparative docking study, at least with the ligands that showed the best results in in vitro assays. In this way, the reasons why compound 6c has a better activity than its triazole congeners can be better supported. In my opinion, this missing information should be included.
- Pharmacokinetics and drug-likeness of triazole 6c should be evaluated using, for instance, a free web tool such as SwissADME.
- In Materials and Methods (Section 3) the computational methodological information followed for the development of the molecular docking study must be included.
Some aspects of form:
The title of Section 1 is missing. It must be included.
In the legend of Figure 2, the meaning of the acronyms FCZ and VOC of the reference drugs must be included. The same for the acronym Mock in the legend of Figure 4.
In line 48, at the end of the sentence "....which have been proved to bind the heme iron and the hydrophobic pocket of CYP51", the respective bibliographic citation must be included.
The period after the word CYP51 in line 189 must be deleted.
In line 232 change “nitrogen” for “nitrogen atmosphere”
Lines 276-277, the spectroscopic procedure followed for the determination of growth inhibition should be better specified.
In line 269, the bibliographic citation referring to the NCCLS standard protocols for in vitro antifungal activity should be included.
Reviewer 3 Report
The manuscript entitled "Design, synthesis, and in vitro and in vivo antifungal activity of novel triazoles containing phenylethynyl pyrazole side chains" presents synthesis and antifungal activity of new triazoles derivatives.
The presented manuscript is fine, but before publication it should be improved:
- Authors write in the "Introduction" about azoles as antifungal agents, but in Fig. 1 are shown only triazoles derivatives. If there are known any azoles containing one or two nitrogen atoms with antifungal activity they should be mentioned in the introduction.
- Figure 2 should be deleted and only the lead structure should be moved to Fig. 1. The corresponding paper with antifungal activity should be cited.
- Line 66: it is not clear the use of word "receptors" in this sentence. Authors should clarify it.
- Fig. 3: "flexible amide/ether as H-bond receptors". It should be "acceptors" instead of "receptors". Furthermore, Authors have introduced "R1" for two series 5 and 6, while it appears from the rest of manuscript that these are different moieties. Moreover, the presented structure for series 5 is not correct: not all compounds contain hydrogen atom at nitrogen atom.
- In Scheme 1 Authors have introduced "R1" for series 6 and "R2" an "R3" for series 5, while in Table 1 the "R". Authors should corrected Fig. 3, Scheme 1 and Tab. 1 and introduced unified designations.
- According to European Committee for Antimicrobial Susceptibility Testing, the MIC value should be reported in "mg/L" instead of "μg/mL". Authors should correct it.
- The results of antifungal activity in Tab. 1 and 2 are presented as MIC values, but in the text (lines 136-138) Authors write MIC80. Authors should clarify this inaccuracy. The antifungal activity should be presented as MIC value, not MIC80.
- Authors should introduced reference concentration ranges, when activity is regarded as excellent, good, moderate etc.
- There 1H NMR analysis should be improved, each of hydrogen atoms being at a specific position should be assigned to observed signal(s).
